# Unconditioned and conditioned anxiolytic effects of Sodium Valproate on flavor neophobia and fear conditioning

**María Ángeles Cintado**[1☯], **Gabriel González**[2☯], **Lucía Cárcel**[1☯], **Luis Gonzalo De la Casa**[1☯] *

1 Laboratory of Animal Behavior & Neuroscience, Dpt. Experimental Psychology, Seville University, Seville, Spain, 2 Jaen University, Jaén, Spain

☯ These authors contributed equally to this work.
* delacasa@us.es

**Data Availability Statement:** All relevant data are within the paper and its Supporting information files.

## Abstract

In three experiments with rats, we analyzed the potential anxiolytic effects of sodium valproate, an anticonvulsant drug that has shown additional pharmacodynamic effects in animal models, including anxiolytic action. Since previous results have revealed that injecting valproate before allowing animals to consume a novel flavor solution resulted in an attenuation of neophobia, we predicted a similar effect when the novel flavor is presented on a drug-free trial in the presence of a context previously associated with the drug. In line with this hypothesis, in our first experiment we observed a reduction in neophobia to a novel flavor for those animals tested in the presence of the context associated with Sodium Valproate. However, a control group that received the drug before being allowed access to the novel flavor showed a significant reduction in consumption. Experiment 2 revealed that the unconditioned effects of the drug include a deleterious effect on the animals' locomotor activity that probably interferes with drinking behavior. Finally, in a third experiment, we directly tested the potential anxiolytic properties of sodium valproate by injecting the drug before implementing a fear conditioning procedure. These findings are explained in terms of the unconditioned anxiolytic action of the drug and the formation of an association between the context and the effects of the drug that evokes a conditioned response reminiscent of such anxiolytic effect.

## Introduction

Pavlov [1] described some experiments to show that repeated morphine administration in the presence of an experimental context resulted in a conditioned response to that context that mimicked some of the unconditional properties of morphine, such as nausea, vomiting, excessive salivation, or sleeping. Subsequently, ample experimental evidence has demonstrated conditioning using the effects of many different drugs as the Unconditioned Stimulus (US) [2–5]. These studies have focused not only on analyzing the learned associations that emerge after pairing the Conditioned Stimulus (CS) and the US [6–8], while also identifying the structures

**Funding:** This research was funded by Agencia Estatal de Investigación (AEI) of Spain (grant no.: PID2019-107530GB-I00/AEI/10.13039/501100011033). The funders had no role in study design, data collection and analysis, decision to publish, or preparation of the manuscript.

**Competing interests:** The authors have declared that no competing interests exist.

and neural circuits that constitute the neurobiological bases of learning [9–12], and the causes of drug addictions since conditioning has been proposed as a potential trigger of relapse and tolerance [13–15].

In three experiments, we explored the result of pairing a neutral stimulus (a distinctive context) with the effects of Sodium Valproate (or Valproic Acid, VPA), one of the most widely used and effective anticonvulsants for the treatment of epilepsy [16]. The main action of VPA on the nervous system is related to the levels of gamma-aminobutyric acid (GABA), and its antiepileptic properties are largely based on the increased levels of this neurotransmitter after its administration [17, 18]. It is this agonist effect on the GABAergic system that underpins its potential anxiolytic properties, which have already been observed in several experiments with rodents, showing a reduction in behaviors associated with anxiety after VPA administration [19–22]. Thus, for example, Corbett et al. [23] employed various tests to measure anxiety in rodents treated with VPA (Geller test, cross maze, or social interaction tests) and observed that, after systematic administration, the animals showed a reduction in anxiety-related behaviors. In a similar vein, Kinrys et al. [24] studied the effects of VPA in people with a social anxiety disorder. After a 12-week treatment with the drug, they observed a significant reduction in symptoms, suggesting its potential use for treating this type of disorder.

In a study aimed at exploring the potential anxiolytic properties of VPA, Shepard [25] analyzed the effects of administering various doses of the drug (100 and 300 mg/kg) on flavor neophobia (using a compound of citric acid and sodium saccharin dissolved in water). Flavor neophobia is a defensive mechanism that produces a reduction in consumption of a new flavor [26], and seems to be related, at least to some degree, to anxiety [27, 28]. In terms of survival, this mechanism protects the organism against potentially dangerous foods since consumption is reduced, thereby limiting any possible poisonous consequences. Alternatively, if an aversive effect does not follow consumption, the initial rejection of the flavor disappears, and it becomes a safe or familiar stimulus due to habituation of neophobia, which results in a progressive increase in flavor consumption [29].

The results of the neophobia test in the experiment conducted by Shepard [25] showed that administering VPA before allowing the animals to consume the novel flavor resulted in increased consumption (i.e., attenuation of neophobia) compared to a control group that received a vehicle. Shepard [25] interpreted these results in terms of the potential anxiolytic action of VPA that would have reduced the anxiety induced by encountering the novel flavor.

In this paper, we analyzed the potential anxiolytic properties of VPA (300 mg/Kg) and the possibility that such properties could be associated with a neutral context so that, later on, such a context elicits a conditioned response that mimics the anxiolytic properties of the drug. More specifically, we wanted to confirm whether the context-CS elicited an anxiolytic conditioned response that reduced the intensity of neophobia reported by Shepard [25]. To this end, in the first experiment we presented a novel flavor in the presence of a context that had been repeatedly associated with VPA. The second experiment was designed to replicate the unconditioned effect of VPA on neophobia and to detect potential alterations in locomotor activity as a side effect of the drug. Finally, a third experiment was designed to check the anxiolytic properties of valproate on fear conditioning, a procedure highly dependent on emotional factors [30, 31].

We anticipated that repeated pairings of a distinctive context with the effects of the drug would establish the context as a CS capable of evoking an anxiolytic conditioned response. Consequently, neophobia to a novel flavor would be reduced when the animal is tested in the presence of the CS-context. Additionally, we expected that the unconditioned anxiolytic properties of valproate would reduce neophobia and fear-conditioned intensity.

## Experiment 1

The main objective of this experiment was to test whether the repeated pairing of a context (the CS) with the anxiolytic effects derived from the administration of VPA (the US) induces a conditioned response to the context that reduces the intensity of neophobia (and its habituation) in the same way as when the drug is directly administered prior to consumption of the novel flavor [25]. However, such an effect would not be observed when VPA is administered after context exposure or in a saline-treated group. Finally, we also expect to replicate the attenuation of neophobia reported by Shepard [25] when the drug is injected before consumption of the novel flavor.

### Method

**Subjects.** Thirty-two experimentally naïve male Wistar rats (group size n = 8), participated in this experiment. The mean weight at the start of the experiment was 356 g. At the arrival to the laboratory, the animals were housed in groups of 2/3 (depending of the animals' weight) in type IIIH cages (820 cm2), with wood savings as bedding, and other materials available in the cages (pieces of fabric, cardboard and wood, stones, etc.), except for the time they were submitted to the experimental procedure when they were individually housed. The vivarium was maintained on a 12:12 h light-dark cycle (lights on at 07:00 h), and all behavioral testing was conducted during the light period of the cycle. Four days before the start of the experimental sessions, each animal was handled daily for 5 min. The day before to initiate the experimental treatment all animals were placed on a water deprivation schedule (30 min/day access to water) that was maintained across the entire duration of the experiment. All experimental procedures were approved by the Ethics Committee for Animal Research, University of Seville (Protocol CEEA-US2020-15/2), and were conducted in accordance with the guidelines established by the EU Directive 2010/63/EU for animal experiments, and the Spanish R. D. 223/1988.

**Apparatus and drugs.** All experimental sessions were conducted in eight Plexiglas cages, 21 x 18 x 35.5 cm, with paper strips as bedding, located in an experimental room, different to the vivarium, illuminated by a single 54-W fluorescent white light on the ceiling. All fluids were provided at room temperature in 150 ml graduated plastic bottles, fitted with stainless steel spouts. The bottles were attached to the front of each cage during each trial. The amount of fluid intake was measured by calculating the difference between the weight of the bottle before and after fluid presentations. Tap water was available in the bottles during the context habituation stage, and a solution of 0,04% sodium saccharine and 0,1% citric acid solution dissolved in tap water was the fluid presented during neophobia trials.

The drug injected was Sodium Valproate (Merck LifeScience), administered intraperitoneally at a dose of 300 mg/kg, 30 min before or after starting each experimental session, depending on the group. A saline solution was used as vehicle.

**Procedure.** A summary of the experimental treatment appears in Table 1. Each animal was randomly assigned to one of the following groups: Sal-Sal/Sal, VPA-Sal/Sal, Sal-VPA/Sal, Sal-Sal/VPA, and, where the first term is referred to the substance injected 30 m before context exposure in the conditioning stage, the second term to the substance injected 30 m after context exposure in the conditioning stage, and the third term to the substance injected immediately before neophobia trials.

The context-conditioning phase took place from day 1 to 4, and it consisted of one daily session during four consecutive days. On these sessions, each animal was introduced in the cages located in the experimental room where they remained for 60-min with access to water in the same bottles that were used in the neophobia stage. Those rats in the VPA-Sal/Sal

**Table 1. Design of Experiment 1.**

| GROUP | Context Conditioning (Days 1–4) | Neophobia Test (Days 7–9) |
|---|---|---|
| Sal–Sal / Sal | Sal– 60 m context (water)—Sal | Sal– 30 min context (flavor) |
| VPA–Sal / Sal | Val– 60 m context (water)- Sal | Sal– 30 min context (flavor) |
| Sal–VPA / Sal | Sal– 60 m context (water)- VPA | Sal– 30 min context (flavor) |
| Sal–Sal / VPA | Sal– 60 m context (water)- Sal | VPA– 30 min context (flavor) |

VPA: 300 mg/Kg of Sodium Valproate; Sal: Saline Solution; Flavor: 0,04% sodium saccharine and 0,1% citric acid dissolved in tap water. See text for additional details.

Group received an i.p injection of Sodium Valproate 30 min before being introduced into the experimental context, and the vehicle 30 min after being returned to their home cages; those rats in the Sal-VPA/Sal condition received the vehicle 30 m before experimental context exposure, and the drug 30 m after they were removed from the experimental chamber. Finally, the Sal-Sal/VPA and the Sal-Sal/Sal groups received the vehicle injections using the same time schedule described for the remaining groups. Mean water consumption for each trial was registered after 30 and 60 min. The animals received an additional 30-min period of water access in the home cages after the context-exposure period.

On Days 5 and 6 the animals remained undisturbed in their home cages where they received the corresponding 30-min period of access to water.

From Days 7 to 10 those animals in the VPA-Sal/Sal, the Sal-VPA/Sal, and the Sal-Sal/Sal groups received an i.p. injection of the Vehicle 30 min before to be introduced in the experimental context, and those rats in the Sal-Sal/VPA Group were injected with sodium valproate. Each one of the three experimental trials consisted in 30-min access to the citric acid + saccharin solution in the experimental cages to evaluate the intensity of neophobia and the subsequent habituation of this response. Ml. consumed were registered as an index of neophobia. An additional 30-min period of access to water was available in the home cages after each experimental session to ensure an appropriate level of hydration for the animals.

## Results

**Water consumption during context conditioning stage.** Fig 1 depicts mean water consumption across the four days of context conditioning after 30 min (Section A), and total consumption for each 60 min trial (Section B). As can be seen in the Fig 1A, when consumption was registered after 30 min, it was lower for the animals in the VPA-Sal/Sal Group and tended to be reduced across days for the Sal-VPA/Sal Group. Those groups that did not receive VPA at conditioning stage (Sal-Sal/VPA, and Sal-Sal/Sal) drank a high and steady amount of water across trials. However, as depicted in Fig 1B, the differences between groups disappeared when considering the total duration of each 60-min trial, suggesting that any effect of the drug on consumption was limited to the first half of each trial.

These impressions were confirmed for the statistical analyses. A 4 x 4 mixed ANOVA (Trials x Groups) conducted on mean amount of water consumed **during the first 30 min** on each conditioning trial revealed significant main effects of Trials and Groups, $F_{(3, 84)} = 4.49$; $p < .01$, $\eta^2 = .14$, and $F_{(3, 28)} = 5.95$; $p < .01$, $\eta^2 = .39$, respectively. The two-way interaction was non-significant, $F_{(9, 84)} = 1.24$; $p < .28$. The main effect of Trials was due to a progressive reduction in consumption across trials. Post-hoc comparisons between groups (Bonferroni, $p < .05$) revealed that mean water consumption for group VPA-Sal/Sal (Mean = 7.51 ml, SD = 1.21) was reduced as compared to Groups Sal-Sal/VPA, and Sal-Sal/Sal (Mean = 9.54 ml,

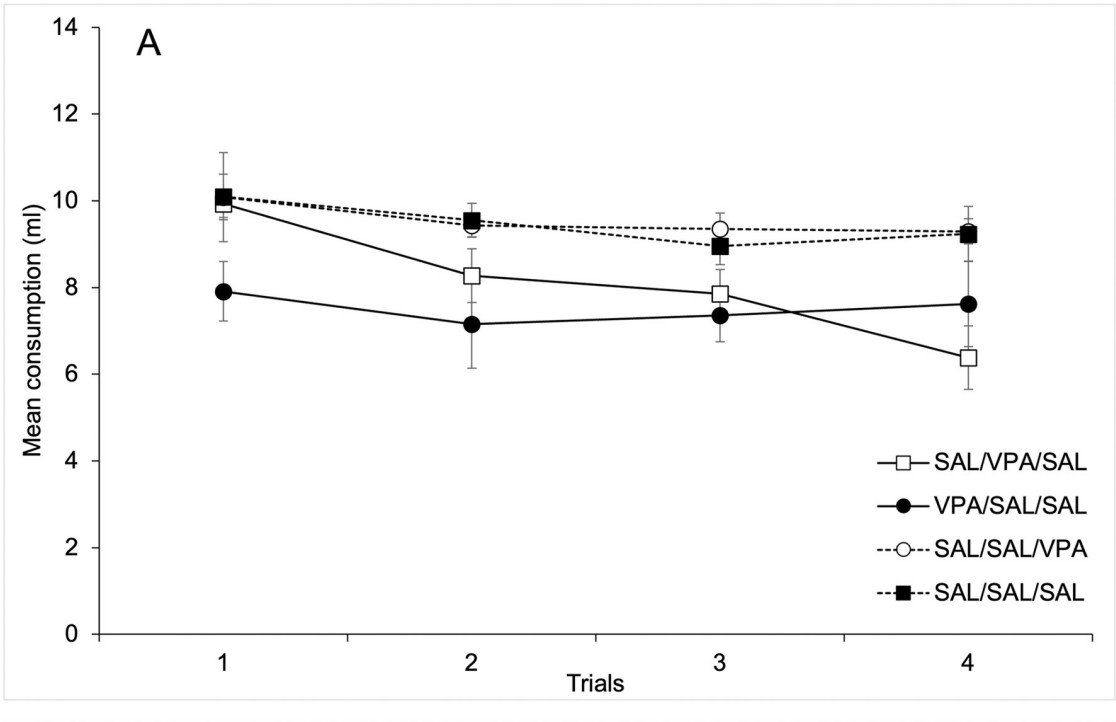

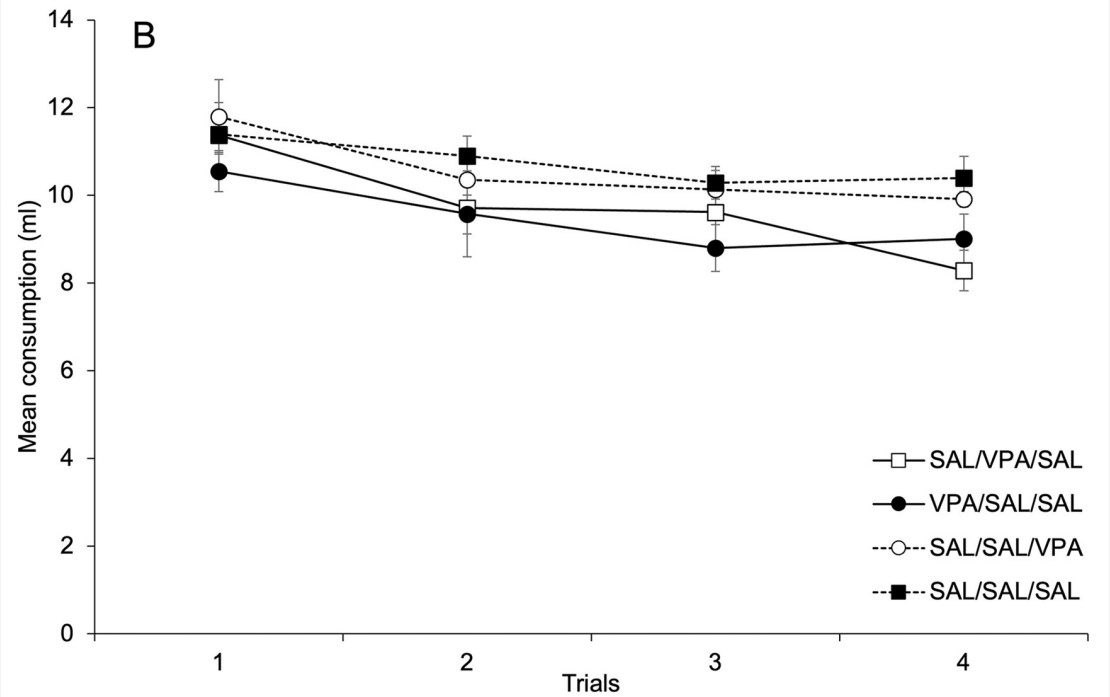

**Fig 1. Mean water consumption across the four days of context conditioning after 30 min (Section A), and for the complete period of 60 min (Section B).**

SD = 0.07, and Mean = 9.46 ml, SD = 1.15, respectively). No more comparisons between groups were significant.

A similar 4 x 4 mixed ANOVA (Trials x Groups) conducted on mean amount of water **consumed across the entire 60 min period** showed a significant main effect of Trials, $F_{(3, 84)} =$

11.61; p < .001, $\eta^2$ = .29, confirming the progressive reduction in consumption across trials that has been observed after 30 min. Neither the main effect of Groups nor the 2-way interaction was significant, $F(3, 28)$ = 2.57; p>.07, $\eta^2$ = .29, and $F(9, 84)<1$.

**Citric acid + saccharine consumption at neophobia stage.** Fig 2 depicts mean citric acid + saccharine consumption across the three trials of the neophobia stage as a function of Groups. As can be seen in the figure, the lower consumption corresponded to the animals in the Sal-Sal/VPA Group (that received an injection of 300 mg/Kg of Valproate 30 min before each neophobia trial). The higher level of consumption appeared in the VPA-Sal/Sal Group, while the Sal-VPA/Sal, and the Sal-Sal/Sal Groups drank an intermediate amount of the flavored solution.

A 3 x 4 mixed ANOVA (Trials x Groups) conducted on mean amount of flavored solution consumed for neophobia trials revealed that both the main effect of Trials and Groups were significant, $F(2, 56)$ = 54.51; p < .001, $\eta^2$ = .66, and $F(3, 28)$ = 49.36; p < .001, $\eta^2$ = .84, respectively. The main effect of Trials reflects an overall increase in consumption across trials due to neophobia habituation. Post-hoc comparisons between groups (Bonferroni, p < .05) on mean consumption collapsed across test trials revealed that consumption in Group VPA-Sal/Sal was significantly higher as compared to Sal-VPA/Sal and Sal-Sal/Sal groups. In addition, consumption for the Sal-Sal/VPA group was reduced as compared to the remaining groups. No more differences were significant.

The Trials x Group interaction was also significant, $F(6, 56)$ = 13.04; p < .001, $\eta^2$ = .58. In order to identify the source of the interaction, we conducted post-hoc comparisons between

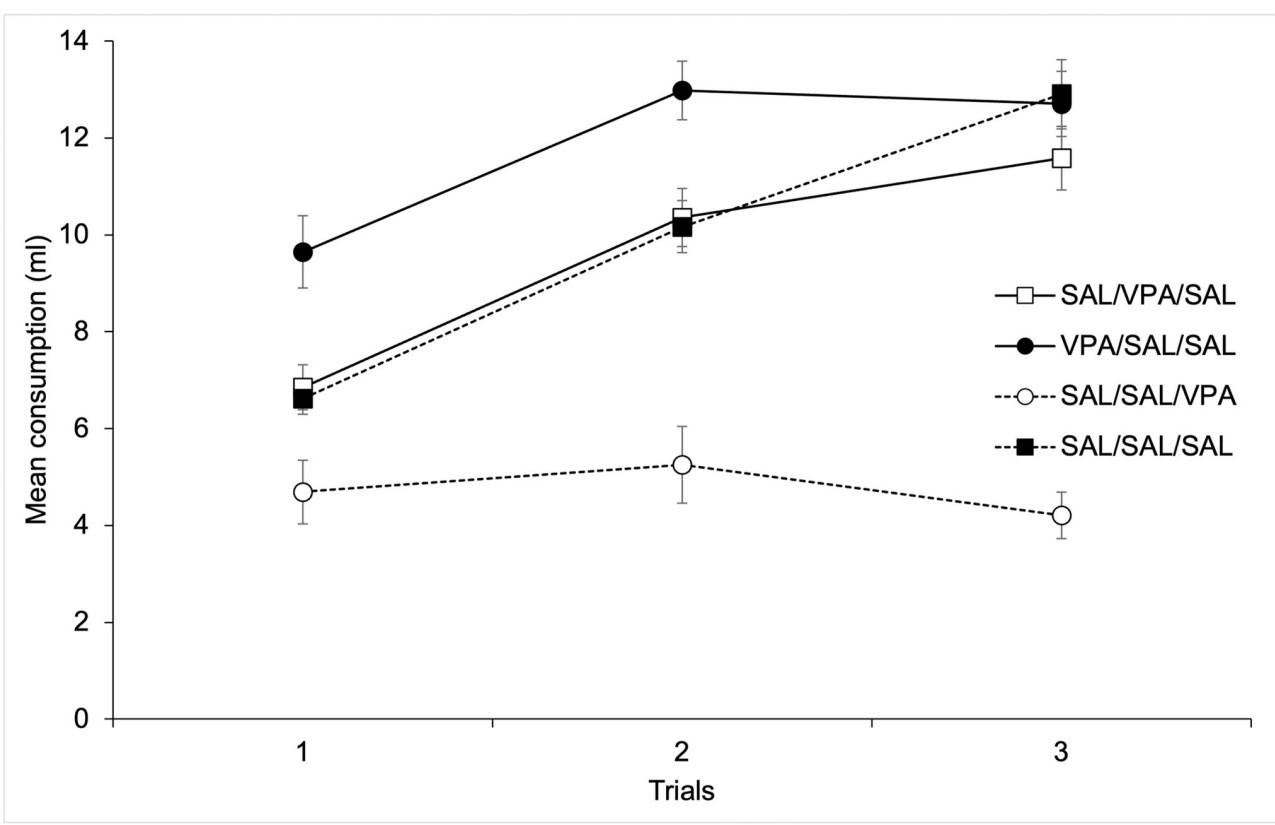

**Fig 2. Mean citric acid + saccharine consumption across the three trials of the neophobia stage as a function of Groups.**

groups (Bonferroni, p < .05) for each Trial that showed higher consumption in the VPA-Sal/Sal group and reduced consumption in the Sal-Sal/VPA group as compared to the remaining groups for the first and second test trials. On the third test trial the Sal-Sal-VPA group drank significantly less than the remaining groups. No more differences between groups were significant.

## Experiment 2

Our hypotheses were partially confirmed in Experiment 1 since neophobia and its habituation were reduced, i.e., consumption of the novel flavor was increased for the VPA-Sal/Sal Group for which the flavor was presented in a context previously paired with the effects of VPA. However, the reduced consumption of the novel flavored solution for the Sal-Sal/VPA Group at the neophobia stage was unexpected since, using the same parameters, Shepard [25] observed the opposite result, namely an increase in consumption of the novel solution when 300 mg/Kg of VPA was i.p. injected 30 m before access to the solution. Similarly, water consumption was reduced for the group that received VPA before the conditioning trials, but this reduction was only evident during the first 30 minutes of each trial.

A possible reason for the reduced consumption observed after VPA administration could be related to the motor disturbances induced by VPA that have been reported in previous studies. In fact, the dose injected in our experiments (300 mg/Kg) is above those that have demonstrated anticonvulsive effects in rats and can induce side effects such ataxia, hypolocomotion, reduction of grooming, muscle relaxation, or wet dog shake behavior [32–35]. We did not register such type of behavior, but can anticipate that these disturbances could have disrupted normal drinking behavior, resulting in reduced consumption.

This possibility was tested in the second experiment, which included three groups exposed to the same novel solution employed in the first experiment. One group received an i.p. injection of 300 mg/kg 30 min before flavor exposure (Group VPA-30), the second group received VPA just before allowing the animal to drink the novel flavor (Group VPA-0), while half of the animals in the third group received saline 30 min before flavor exposure, and the other half saline immediately before such exposure (Group Sal).

Neophobia tests were conducted for three consecutive days (30 min access to the flavor each day) in experimental chambers designed to detect the animals' movements. Based on the results from Experiment 1, we anticipated a reduction in drinking for those animals in the VPA groups, which could be even greater for the VPA-0 compared with the VPA-30 Group. Additionally, we expected a reduction in the global percentage of locomotor activity for the animals that received VPA compared to those that received saline.

### Method

**Subjects.**  Twenty-four experimentally naïve male Wistar rats (group size n = 8), participated in this experiment. The mean weight at the start of the experiment was 355 g. The animals were housed and maintained exactly as described for Experiment 1.

**Apparatus and drugs.**  All experimental sessions were conducted in four identical Panlab conditioning boxes (model LE111, Panlab/Harvard Apparatus, Spain), each measuring 26 x 25 x 25 cm (H x L x W). Each chamber was enclosed in a sound-attenuating cubicle (model LE116. Panlab/Harvard Apparatus, Spain). The walls of the experimental chambers were made of white acrylic, and the floor consisted of stainless steel rods, 2 mm in diameter, spaced 10 mm apart (center to center). Each chamber rested on a platform that recorded the signal generated by the animal movement through a high sensitivity Weight Transducer system. Such signal was automatically converted into percent of general activity, defined as the percentage

of the total time that movement was detected on 2-min periods, by a commercial software (StartFear system software, Panlab/Harvard Apparatus, Spain). Sampling was performed continuously at a frequency of 50Hz. All fluids were provided at room temperature in 150 ml graduated plastic bottles containing a 0,04% sodium saccharine and 0,1% citric acid solution dissolved in tap water that were attached to the front of each cage during each trial. As in Experiment 1, the amount of fluid intake was measured by calculating the difference between the weight of the bottle before and after fluid presentations.

The drug injected was Sodium Valproate (Merck LifeScience), that was i.p. administered at a dose of 300 mg/kg 30 min before (Group VPA/30) o immediately before (Group VPA/0) to start each neophobia session. An additional control group (Group Sal) received only the saline solution. For this control group half of the rats received the saline injection 30 min before the neophobia trial, and the other half received the injection immediately before of each trial.

**Procedure.** Two days before to start the experimental treatment the standard bottles used in the vivarium were replaced for the bottles employed during the experimental trials containing water, in order to habituate the animals to them.

The experimental treatment lasted three days, and each day the animals has access to 30 min of access to the citric acid + saccharin solution. Those animals in the VPA/0 and the VPA/30 groups received an i.p. injection of VPA immediately or 30 min before, respectively, to the start of each experimental trial, that consisted in 30-min access to the citric acid + saccharine solution. Mean percent of activity and consumption were registered. An additional 30-min period of access to water was available at the home cages after each experimental session to ensure an appropriate level of hydration for the animals.

## Results

A preliminary comparison showed no significant differences neither in consumption nor locomotor activity level between those animals that received the saline injection 30 or 0 min before novel flavor exposure. Therefore, the data for all the animals in the Sal Group were unified for the statistical analyses.

**Citric acid + saccharine consumption.** Fig 3 depicts mean citric acid + saccharine consumption for the three neophobia trials. As can be seen in the figure, the expected effect of neophobia appeared in the Sal Group at the first trial, and it was followed by neophobia habituation for second and third trials. Conversely, the animals in the VPA/30 and VPA/0 groups showed a reduced amount of drinking across trials that was even more intense for the VPA/0 group.

A 3 x 3 mixed ANOVA (Trials x Groups) confirmed these impressions. The analyses revealed significant main effects of Trials and Groups, $F_{(2, 42)} = 4.3$; $p < .05$, $\eta^2 = .17$, and $F_{(1, 21)} = 14.48$; $p < .001$, $\eta^2 = .58$, respectively. The main effect of Trials was due to an overall increase in consumption across trials. Post hoc comparisons (Bonferroni, $p < .05$) revealed that the effect of Groups was due to higher mean consumption for the Sal (Mean = 7.72 ml, SD = 5.05) as compared to both VPA/30 and VPA/0 groups (Mean = 2.87 ml, SD = 2.32, and Mean = .61 ml, SD = .43, respectively). The Trials x Groups interaction was also significant, $F_{(4, 42)} = 7.63$; $p < .001$, $\eta^2 = .42$. Post-hoc comparisons between groups (Bonferroni, $p < .05$) for each trial showed lower consumption in the VPA/0 group as compared to the VPA/30 for the first trial and to the Sal Group for all trials. Consumption in the VPA/30 Group was also reduced as compared to Sal Group for the third trial. No more differences between groups were significant.

**Mean percent activity.** Activity percent was computed for each one of the three trials in five periods of 6 minutes. A 3 x 5 x 3 mixed ANOVA, with main factors Trials, 6-min periods,

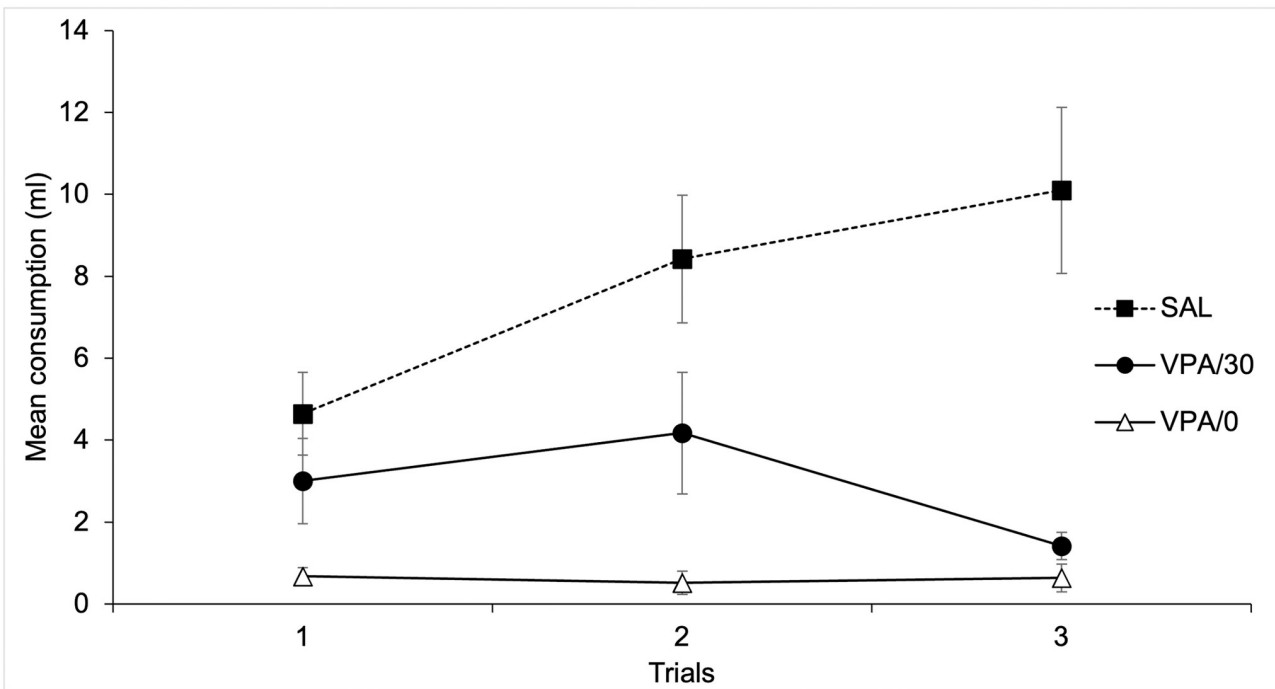

**Fig 3. Mean citric acid + saccharine consumption for the three neophobia trials as a function of Groups.**

both within subjects, and Groups revealed a significant main effect of Periods and Groups, $F(4, 168) = 64.93$; $p < .001$, $\eta^2 = .76$ y $F(2, 21) = 41.66$; $p < .001$, $\eta^2 = .79$, respectively. The main effect of Periods was due to a progressive decrease in activity across time. Post-hoc comparisons between groups (Bonferroni, $p < .05$) revealed that the main effect of Groups was due to a higher global level of activity for the Sal (Mean = 69.46%, SD = 4.14) as compared to the VPA/0 and the VPA/30 groups (Mean = 30.10%, SD = 2.69, and Mean = 31.41%, SD = 3.41, respectively).

The Periods x Groups interaction was also significant, $F(8, 168) = 8.68$; $p < .001$, $\eta^2 = .45$. No more main effects or interactions were significant (all ps>.58). The Periods x Groups interaction is depicted in Fig 4, that shows mean percent activity for the 6-min periods collapsed across the three test sessions as a function of Groups. As can be seen in the figure, and confirmed by post hoc comparisons between groups (Bonferroni, $p < .05$), mean activity was lower for the VPA/0 as compared to the VPA/30 Group for the first 6-min period, and for all periods as compared to the Sal Group. Mean activity in the VPA/30 Group was significantly reduced as compared to the Sal Group from the second to the sixth 6 m-period.

## Experiment 3

Experiment 2 confirmed disturbances in locomotor activity induced by VPA administration that could have interfered with drinking behavior. More specifically, a reduction in movement can impair the animal's ability to approach the tubes, consequently reducing the amount of solution consumed. However, we do not have direct evidence of the potential anxiolytic effect of VPA that might explain the reduction of neophobia and its habituation observed in Experiment 1 when consumption took place in the presence of the context associated with the drug.

The third experiment was designed to obtain a direct behavioral test of the anxiolytic effect of VPA using a fear-conditioning paradigm, a process that has been linked to emotional

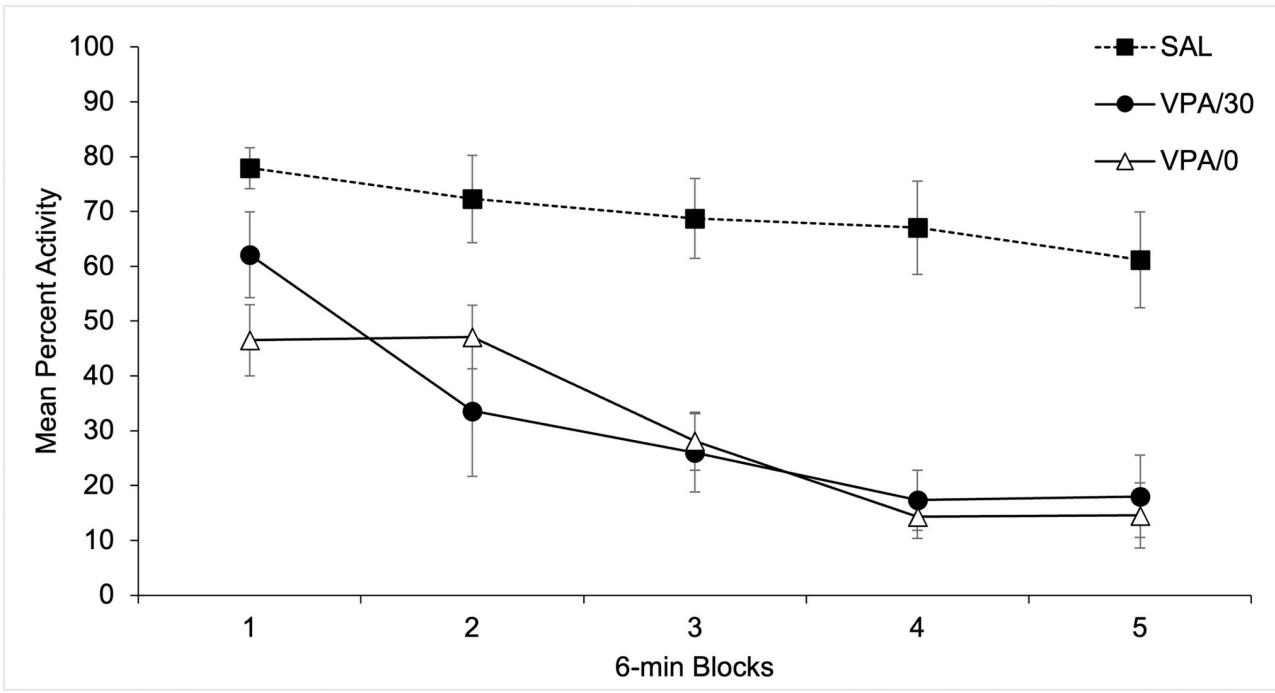

**Fig 4. Mean percent activity for the 6-min periods collapsed across the three test sessions as a function of Groups.**

factors [30, 31]. To this end, we designed an experiment to determine whether VPA can reduce fear-conditioning intensity. More specifically, the experiment comprised three groups: a VPA/ Before Group received VPA before presenting 3 tone-shock pairings; a VPA/After Group received the drug after conditioning trials; and a Sal/Before Group received a saline injection before conditioning. Next, the day after conditioning, the tone was repeatedly presented without the shock in a drug-free session to evaluate the intensity of fear conditioning.

Given the potential anxiolytic effect of VPA, we expected to find reduced fear conditioning in the group that received VPA before conditioning as compared to the groups that received saline or VPA after the tone-shock pairings.

## Method

**Subjects.**   Twenty-four experimentally naïve male Wistar rats (group size n = 8), participated in this experiment. The mean weight at the start of the experiment was 423 g. The animals were housed and maintained exactly as described for Experiment 1, except for water that was available *ad libitum*.

**Procedure and apparatus.**   All experimental sessions were conducted in the same conditioning boxes described in Experiment 2. The conditioning session started with a 180 s period without any stimulation followed by 3 tone-shock pairings with a 180 s Inter Trial Interval. The US was a 1-s, 0.5-mA unscrambled AC 50-Hz foot shock from a constant-current generator (Model LE100-26) that was delivered to the floor of each chamber. A loudspeaker was located at the top of each chamber, which produced a 70 dB 2.8-kHz 30 s tone that was used as CS. For conditioning session, an i.p. injection of VPA (300 mg/kg) was injected 30 min before (Group VPA/Before) o immediately after (Group VPA/After) the start of the experimental treatment. An additional control group (Group Sal/Before) received a vehicle 30 min before

the start of the conditioning stage. One animal from the VPA/After Group was removed from the experiment due to a failure in an experimental chamber during the conditioning stage.

The next day, a drug-free extinction session similar for all groups was conducted. It started with a 180 s period without any stimulation followed by six tone-alone presentations with a 180 s Inter Trial Interval. The chambers' floor rested on a platform that registered and recorded the animal's movements. A percentage score indicating freezing was computed by the experimental software (PANLAB Startfear) for the proportion of the total time that immobility was detected.

## Results

We conducted preliminary one-way ANOVAs on mean percent freezing during the 180 s period prior to the onset of the first CS-tone for conditioning and testing stages with Groups as main factor. The analyses for the conditioning stage revealed a significant effect of Groups, $F(2, 20) = 47.63$; $p < .001$, $\eta^2 = .82$. Post-hoc comparisons between groups (Bonferroni, $p < .05$) revealed a higher level of freezing for the VPA/Before (Mean = 87.42%, SD = 27.03) as compared to the VPA/After and Sal/Before groups (Mean = 13.88%, SD = 3.84, and Mean = 19.37%, SD = 6.34). This result confirms the reduction in locomotor activity induced by VPA.

The analyses for the first pre-CS period for the extinction stage also revealed differences between Groups, $F(2, 20) = 17.03$; $p < .001$, $\eta^2 = .63$. Post-hoc comparisons between groups (Bonferroni, $p < .05$) revealed a lower level of freezing for the VPA/Before (Mean = 30.91%, SD = 23.84) as compared to the VPA/After and Sal/Before groups (Mean = 87.71%, SD = 12.88, and Mean = 81.61%, SD = 20.92). These differences probably indicate that fear conditioning generalized to the context, but to a lesser extent in the group that received VPA before conditioning stage.

Fig 5 depicts mean freezing to the Tone-Cs during conditioning (left side) and extinction (right side) sessions. As can be seen in the figure, activity for the VPA/Before Group was reduced during the first conditioning trial (i.e., freezing was higher) as compared to the remaining groups, replicating the deleterious effect of the drug on locomotor activity observed in Experiment 2. As for the VPA/after and Sal/Before groups, conditioning was expressed through a high level of freezing in presence of the tone-CS for the second and third conditioning trials. Regarding the drug-free extinction stage, fear conditioning was more intense for VPA/After and Sal/Before groups, as revealed for the higher levels of freezing in presence of the tone during the first extinction trials for these groups as compared to the VPA/Before Group.

A 3 x 3 mixed ANOVA (Trials x Groups) conducted on mean percent of freezing during conditioning revealed a significant main effect of Trials and Groups, $F(2, 40) = 39.28$; $p < .001$, $\eta^2 = .66$ y $F(2, 20) = 4.45$; $p < .05$, $\eta^2 = .31$. The main effect of Trials reflects a progressive increase in freezing across conditioning trials. Regarding the effect of Groups, post-hoc comparisons (Bonferroni, $p < .05$) revealed a higher global level of freezing for the VPA/Before (Mean = 77.09%, SD = 14.68) as compared to the VPA/After group (Mean = 56.43%, SD = 11.94). No more comparisons were significant. The Trials x Groups interaction was also significant, $F(4, 40) = 4.6$; $p < .01$, $\eta^2 = .32$. In order to explore the source of the interaction we conducted Post-hoc comparisons (Bonferroni, $p < .05$) between groups for each trial that that showed significant differences between the VPA/Before as compared to the Sal/Before and the VPA/After groups.

A 6 x 3 mixed ANOVA (Trials x Groups) conducted on mean percent of freezing during extinction revealed a significant main effect of Groups, $F(2, 20) = 10.70$; $p < .01$, $\eta^2 = .52$. Post-

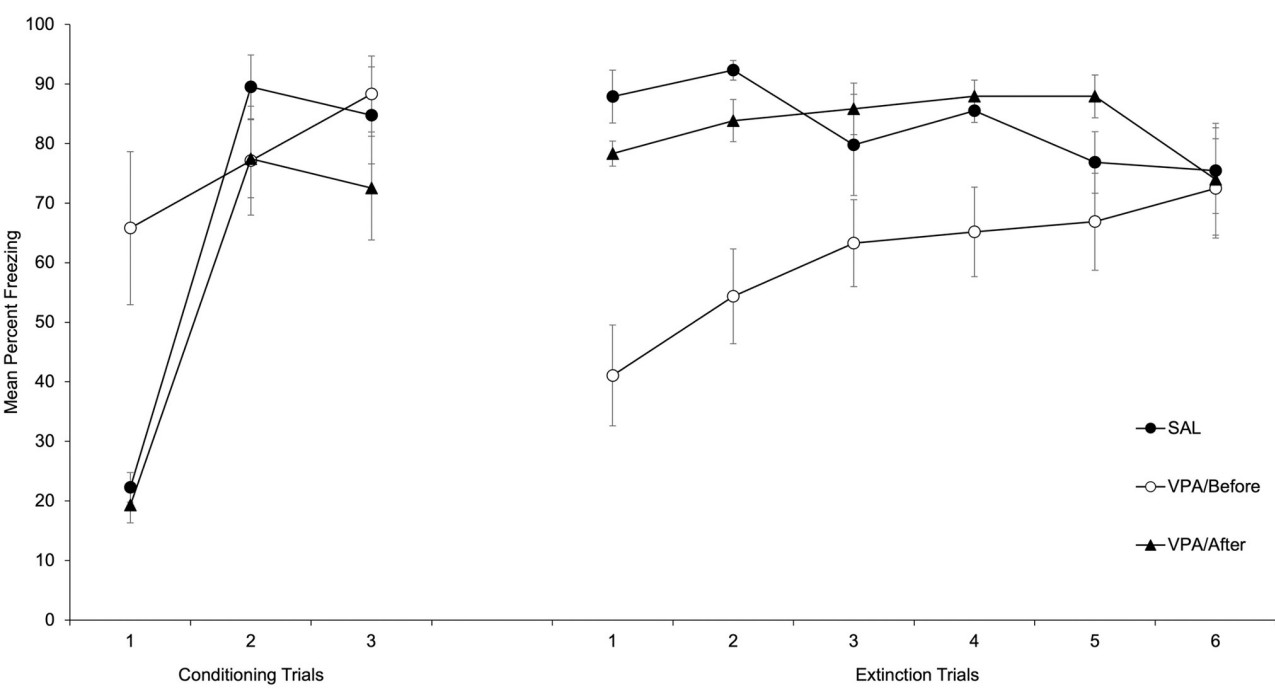

**Fig 5. Mean freezing to the Tone-Cs during conditioning (left side) and extinction (right side) sessions as a function of Groups.**

hoc comparisons between groups (Bonferroni, p < .05) revealed a lower global level of freezing for the VPA/Before (Mean = 60.54%, SD = 13.99) as compared to the VPA/After and the Sal/Before groups (Mean = 82.99%, SD = 10.42, and Mean = 82.96%, SD = 7.89, respectively). The main effect of Trials was non-significant, $F(5, 100) = 1.33$; $p > .26$, $\eta^2 = .06$. The Trials x Groups interaction was also significant, $F(10, 100) = 2.86$; $p < .01$, $\eta^2 = .22$. In order to explore the interaction, we conducted post-hoc comparisons between groups for each trial (Bonferroni, ps < .05) that revealed lower levels of freezing, i.e., less fear conditioning, for the VPA/Before Group as compared to the Sal/Before and VPA/After groups for the 1st, 2nd, and 4th trials. No more comparisons were significant.

## General discussion

The results of this study have revealed that 300 mg/Kg of VPA resulted in both an anxiolytic effect and a reduction of the animals' locomotor activity. Furthermore, the anxiolytic effect was demonstrated by a reduction in fear conditioning in Experiment 3, and as a conditioned response that reduced the intensity of neophobia in Experiment 1. Therefore, the increased consumption of a novel flavor for those rats tested in the presence of a context previously associated with the effect of VPA can only be interpreted in terms of an anxiolytic conditioned response induced by the context-CS; any explanation in terms of residual effects of the drug on the drug-free test day is ruled out by the fact that the animals that received VPA after context exposure showed no change in neophobia. Thus, we were able to confirm our main hypothesis that in this case a Pavlovian association is formed between a distinct context (the CS) and the effects of the drug (the US).

The reduction in fear conditioning for the group that received VPA before the tone-shock pairings in Experiment 3 was probably due to the unconditioned anxiolytic effect of VPA, a property that has been demonstrated in previous research using different anxiety behavioral

task [19–22]. In this case, a possible explanation in terms of lack of memory consolidation of the tone-shock association can be ruled out for the absence of an effect on fear conditioning for the group that received VPA immediately after the conditioning trials. However, we cannot completely rule out an account in terms of the possible effect of the drug on the perceived intensity of the shock, which could have weakened the strength of the association with the tone. However, this possibility is unlikely considering that the level of freezing shown by the VPA-Before group at the end of conditioning treatment was at asymptote and similar to the remaining groups.

The deleterious effect of VPA on locomotor activity was clearly established when the percentage of activity was recorded after drug administration in Experiment 2. The greater detrimental effect of the drug on locomotor activity for those animals that received the drug immediately before the test compared with those that received VPA 30 m before the test, revealed that such an effect was time-dependent and relatively short-lasting. The reduction in locomotor activity is most likely the result of the lower water consumption observed when the animals received VPA before the conditioning stage of Experiment 1 and the reduction in consumption of the novel flavor for the animals injected with the drug in Experiment 2. This result was largely unexpected since Shepard [25] reported an increase in the consumption of a novel flavor after administering both 100 and 300 mg/Kg. of VPA. There are only two apparent procedural differences between Shepard's experiments and those reported here: the context used as the experimental environment and the period for measuring fluid consumption. Regarding the experimental context, Shepard conducted the entire procedure in the animal's home cage. However, the present experiments were run in distinctive cages located outside of the vivarium. This could be a possible factor responsible for the discrepancy in results since it has been demonstrated that consumption of a novel flavor is greater when presented in a familiar and safe home cage than in a novel and potentially dangerous context [36]. A second relevant difference is related to the periods for which intake was recorded since Shepard recorded fluid intake after 6, 18, and 30 m, but we only recorded consumption after 30 m. Importantly, Shepard found a significant increase in consumption after 6 m but not after 18 or 30 m. Therefore, the mentioned procedural discrepancies between experiments could, at least in part, explain the different results, although this possibility merits further investigation. In any case, such differences do not undermine the main conclusions drawn from our experiments, namely the anxiolytic effect of VPA and the emergence of an anxiolytic conditioned response through a classical conditioning process.

Alternative explanations for the changes in neophobia mediated by the anxiolytic conditioned response observed in the VPA-Sal/Sal Group in the test phase of Experiment 1 include the effects of context novelty on fluid consumption [37]. More specifically, considering that VPA administration could have altered context processing during the conditioning stage, it is possible that such a context could have been rendered relatively novel at the time of testing for the animals that received the drug. Furthermore, previous research has revealed that neophobia increases when the flavor is presented in a novel context, possibly due to exploratory responses interfering with consumption [37, 38]. However, in Experiment 1, we observed an increase in consumption for the group that received VPA before context conditioning instead of the reduction that would have been expected if the context had remained functionally novel, thus ruling out an interpretation of our results in terms of exploratory behavior interfering with fluid intake.

The role of context in neophobia has received rather limited attention, possibly because the key factors proposed as determinants of neophobia are mainly related to the characteristics of the novel flavor [39] or to the deprivation state of the animals at the time of consumption [40]. However, the experimental context has shown to play a key role in both neophobia and its

habituation [36]. Thus, in a novel context, the presentation of an unknown flavor produces an increase in neophobia. Consequently, consumption is reduced compared to when the same flavor is presented for the first time in a familiar context [41]. Furthermore, consumption increases when a flavor is presented in contexts previously associated with appetitive stimuli but decreases when the flavor appears in a context associated with aversive stimuli [42]. Similarly, other studies suggest that when a novel taste is presented in a changing environment, the neophobic response is weaker [43]. In the present paper, we have proposed a new role for the context in modulating the consumption of novel flavors since neophobia and its habituation can be reduced when the novel flavor is consumed in the presence of a context previously associated with the anxiolytic effects of a drug.

Regarding the unconditioned anxiolytic effect of VPA, previous studies have proposed that this depends on an elevation of GABA levels [44] in addition to blocking of GABA re-uptake [45]. Moreover, some data indicate that the action of VPA resembles that of benzodiazepines, with which it also shares anticonvulsant properties and acts as a muscle relaxant [46, 47]. Such anxiolytic effects of VPA have been demonstrated with a wide variety of procedures, such as the consumption of new flavors with high palatability [48, 49], different types of conflict tasks both with shock [21] and without shock [19], the light-dark aversion test [20], the staircase test [22], the elevated plus-maze [50], or social interaction [23]. As mentioned above, we have directly confirmed the anxiolytic effect of VPA using both a fear conditioning procedure (Experiment 3), and the effects of a conditioning process on neophobia (Experiment 1).

Recent studies suggest that VPA may be useful for the treatment of a wide range of pathologies, such as bipolar disorder [51–53], migraines [54–56], disorders related to motor disturbances [57, 58], and even in cancer treatment [59–61]. Among these novel and promising applications of VPA, stands out its possible use as an anxiolytic agent [23, 62–64], due to the above mentioned agonistic action on the GABAergic system, similar to the mechanisms of action of benzodiazepines [21, 24, 64, 65].

Thus, it seems that this drug could be useful in a wide range of disorders in which anxiety plays a leading role. For example, there are studies suggesting its effectiveness for the treatment of social anxiety disorders [24], post-traumatic stress disorders [66, 67], and generalized anxiety disorders [68, 69], among others. The findings of our study support previous research on potential application of this drug as an anxiolytic agent and include evidence of the contextual influence that accompanies drug use, already demonstrated in other substances [8, 9, 70, 71], and increasingly present in the clinical treatment of anxiety [72–74].

In sum, we have demonstrated that VPA (300 mg/Kg) induces a reduction in locomotor activity and has anxiolytic properties. Interestingly, this anxiolytic effect can be associated with a neutral stimulus (a distinctive context) and expressed as a conditioned response that reduces the intensity of neophobia and its habituation to a new flavor. These findings have implications for the potential therapeutic use of VPA, our understanding of the mechanisms of classical conditioning using drug stimuli, and the links between contextual cues and neophobia.

## Supporting information

**S1 File. Data for Experiment 1.**
(SAV)

**S2 File. Data for Experiment 2.**
(SAV)

**S3 File. Data for Experiment 3.**
(SAV)

## Author Contributions

**Conceptualization:** María Ángeles Cintado, Gabriel González, Lucía Cárcel, Luis Gonzalo De la Casa.

**Formal analysis:** María Ángeles Cintado, Gabriel González, Lucía Cárcel, Luis Gonzalo De la Casa.

**Funding acquisition:** Luis Gonzalo De la Casa.

**Investigation:** María Ángeles Cintado, Gabriel González, Lucía Cárcel, Luis Gonzalo De la Casa.

**Methodology:** María Ángeles Cintado, Luis Gonzalo De la Casa.

**Project administration:** Luis Gonzalo De la Casa.

**Supervision:** Luis Gonzalo De la Casa.

**Writing – original draft:** María Ángeles Cintado, Gabriel González, Luis Gonzalo De la Casa.

**Writing – review & editing:** María Ángeles Cintado, Gabriel González, Lucía Cárcel, Luis Gonzalo De la Casa.

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
