## [Decision Letter · Decision Letter 0]

9 Mar 2023

PONE-D-22-32967Unconditioned and Conditioned Anxiolytic Effects of Sodium Valproate on Flavor Neophobia and Fear ConditioningPLOS ONE

Dear Dr. De la Casa,

Thank you for submitting your manuscript to PLOS ONE. After careful consideration, we feel that it has merit but does not fully meet PLOS ONE’s publication criteria as it currently stands. Therefore, we invite you to submit a revised version of the manuscript that addresses the points raised during the review process.

ACADEMIC EDITOR: Please do act according to suggestions by our reviewers and also take the additional comments provided by editorial evaluation done to expedite the revision process. 

We look forward to receiving your revised manuscript.

Kind regards,

Dragan Hrncic

Academic Editor

PLOS ONE

Journal Requirements:

This research was funded by Agencia Estatal de Investigación (AEI) of Spain (grant no.: PID2019-107530GB-I00/AEI/10.13039/501100011033)

Additional Editor Comments:

Present manuscript addressed an interesting topic.

Some issues to be ameliorated:

- do not start the sentence with number. i.e. 24 rats.... Also, indicate that group size is denoted as n=8

- avoid repeating common info for each experiment, i.e. Ethical statements. Be clear in declarations and animal welfare committee approval numbers, but once.

- VPA dose of 300 mg/kg is above those achieving anticonvulsive effects in the rats (e.g. PMID: 17714087). Some side effects could occur. Please, discuss. Did you noticed any in general behavior of rats.

- More profoundly discuss the clinical relevance of the findings.

Reviewers' comments:

Reviewer's Responses to Questions

**Comments to the Author**

1. Is the manuscript technically sound, and do the data support the conclusions?

Reviewer #1: Yes

2. Has the statistical analysis been performed appropriately and rigorously? 

Reviewer #1: Yes

3. Have the authors made all data underlying the findings in their manuscript fully available?

Reviewer #1: Yes

4. Is the manuscript presented in an intelligible fashion and written in standard English?

Reviewer #1: Yes

5. Review Comments to the Author

Reviewer #1: Summary: The manuscript reports on three experiments addressing conditioned and unconditioned anxiolytic effects of valproic acid measured in the context of flavor neophobia and auditory fear conditioning. Further, motoric effects of VPA are addressed. The manuscript is well-written. The experiments were rigorously designed and analyzed (except where noted below).

Major concerns:

1) Exp 3: Given the motoric effects of VPA demonstrated in Exp 2, it is especially important to show the freezing levels in the acquisition and extinction phases prior to onset of the first tone. Related to this, was the extinction test conducted in a separate context from the acquisition? If not, how is context freezing distinguished from tone freezing?

2) Exp 3: It would be more appropriate to conduct separate 3 X 3 (Trial X Group) ANOVAs for the acquisition and extinction phases of the experiment, as opposed to the 6 X 3 ANOVA that the authors reported. The two phases are distinct and should be treated as such statistically.

3) Discussion: There are a number of issues concerning the interpretation of Exp 3 results. First, I’m not convinced that the results of Exp 3 say anything about anxiety. It is possible (probable) that VPA administered prior to fear conditioning would have a direct effect on the acquisition of fear, irrespective of any potential effect on anxiety. Second, it is necessary to rule out the (strong) possibility of state-dependent learning effects in Exp 3, especially since VPA has clearly demonstrated unconditioned motoric effects.

4) Generally, I believe this set of experiments, while conducted well, are inconclusive with regards to the overall interpretation of VPA effects on anxiety. Assessment of VPA effects on anxiety using a classic anxiety behavioral task (or multiple tasks) would significantly strengthen the interpretation the authors favor.

Minor concerns:

1) Lines 148-149: List groups in same order (in text) as listed in Table 1.

2) Clarify whether Figure 1B represents total consumption (minutes 1-60) or consumption for minutes 30-60.

3) Line 411: change “free-drug” to “drug-free”

4) It would be useful to include a direct assessment of footshock reactivity during acquisition in Exp 3 (e.g., activity burst; Fanselow, 1982).

Fanselow, Michael S. “The Postshock Activity Burst.” Animal Learning & Behavior, vol. 10, no. 4, Nov. 1982, pp. 448–54. EBSCOhost,

6. PLOS authors have the option to publish the peer review history of their article (what does this mean?). If published, this will include your full peer review and any attached files.

Reviewer #1: **Yes: **Jennifer J. Quinn

---

## [Author Response · Author response to Decision Letter 0]

17 Mar 2023

Responses to each point raised by the editor and the reviewer:

EDITOR:

- do not start the sentence with number. i.e. 24 rats.... Also, indicate that group size is denoted as n=8

Done

 - avoid repeating common info for each experiment, i.e. Ethical statements. Be clear in declarations and animal welfare committee approval numbers, but once.

Done

 - VPA dose of 300 mg/kg is above those achieving anticonvulsive effects in the rats (e.g. PMID: 17714087). Some side effects could occur. Please, discuss. Did you noticed any in general behavior of rats.

The potential side effects of the VPA dose injected in our experiments were mentioned in the introduction of Experiment 2. As for such possible side effects affecting to our sample of animals, we only registered percent time of activity, so we cannot report other specific motoric side effects. Anyway, we have mentioned in the ms that 300 mg/Kg is above the usual anticonvulsive dose and that we did not registered specific movements. The paragraph now reads (p. 12, lines 276-281): “In fact, the dose injected in our experiments (300 mg/Kg) is above those that have demonstrated anticonvulsive effects in rats and can induce side effects such ataxia, hypolocomotion, reduction of grooming, muscle relaxation, or wet dog shake behavior [32,33,34,35]. We did not register such type of behavior, but can anticipate that these disturbances could have disrupted normal drinking behavior, resulting in reduced consumption.”

 - More profoundly discuss the clinical relevance of the findings.

We have included two paragraph in the discussion section describing the potential clinical relevance of our results (p. 24-25, lines 598-614): ”Recent studies suggest that VPA may be useful for the treatment of a wide range of pathologies, such as bipolar disorder [51,52,53], migraines [54,55,56], disorders related to motor disturbances [57,58], and even in cancer treatment [59,60,61]. Among these novel and promising applications of VPA, stands out its possible use as an anxiolytic agent [23, 62, 63, 64], due to the above mentioned agonistic action on the GABAergic system, similar to the mechanisms of action of benzodiazepines [21, 24, 64, 65].

Thus, it seems that this drug could be useful in a wide range of disorders in which anxiety plays a leading role. For example, there are studies suggesting its effectiveness for the treatment of social anxiety disorders [24], post-traumatic stress disorders [66,67], and generalized anxiety disorders (68,69), among others. The findings of our study support previous research on potential application of this drug as an anxiolytic agent and include evidence of the contextual influence that accompanies drug use, already demonstrated in other substances [8,9,70,71], and increasingly present in the clinical treatment of anxiety [72, 73,74]”

REVIEWER #1: 

  Major concerns:

 1) Exp 3: Given the motoric effects of VPA demonstrated in Exp 2, it is especially important to show the freezing levels in the acquisition and extinction phases prior to onset of the first tone. 

We have introduced in the results section of Exp. 3 statistical analyses on freezing prior to the first tone for conditioning and extinction stages. The results revealed a significant increase of freezing for the VPA/Before Group as compare to Sal/Before and VPA/After groups, confirming the reduction of movements induced by VPA. A similar analysis on percent of freezing for the period previous to the first tone presentation in the extinction phase showed a reduce level of freezing for those animals in the VPA/before Group, probably revealing a low level of fear to the context as compared to the control groups. Specifically, the following paragraphs have been introduced in the ms (p. 18-19, lines 446-461): “We conducted preliminary one-way ANOVAs on mean percent freezing during the 180 s period prior to the onset of the first CS-tone for conditioning and testing stages with Groups as main factor. The analyses for the conditioning stage revealed a significant effect of Groups, F(2, 20)=47.63; p<.001, η2=.82. Post-hoc comparisons between groups (Bonferroni, p<.05) revealed a higher level of freezing for the VPA/Before (Mean = 87.42%, SD = 27.03) as compared to the VPA/After and Sal/Before groups (Mean = 13.88%, SD = 3.84, and Mean = 19.37%, SD = 6.34). This result confirms the reduction in locomotor activity induced by VPA. 

The analyses for the first pre-CS period for the extinction stage also revealed differences between Groups, F(2, 20)=17.03; p<.001, η2=.63. Post-hoc comparisons between groups (Bonferroni, p<.05) revealed a lower level of freezing for the VPA/Before (Mean = 30.91%, SD = 23.84) as compared to the VPA/After and Sal/Before groups (Mean = 87.71%, SD = 12.88, and Mean = 81.61%, SD = 20.92). These differences probably indicate that fear conditioning generalized to the context, but to a lesser extent in the group that received VPA before conditioning stage.”

2) Related to this, was the extinction test conducted in a separate context from the acquisition? If not, how is context freezing distinguished from tone freezing?

The acquisition and extinction tests were conducted in the same context. As pointed out by the reviewer, this prevents distinguishing between the freezing produced by the context and that produced by the tone at testing. In fact, as shown in the new analyses included in the results section of Experiment 3, context freezing for the VPA/after and Sal/after groups was significantly higher than that for the VPA/Before group. However, this does not compromise, but rather adds validity to the interpretation proposed in the manuscript, according to which the administration of VPA for the VPA/Before group resulted in a reduction of fear conditioning intensity (both to the Tone-CS and to the context).

 2) Exp 3: It would be more appropriate to conduct separate 3 X 3 (Trial X Group) ANOVAs for the acquisition and extinction phases of the experiment, as opposed to the 6 X 3 ANOVA that the authors reported. The two phases are distinct and should be treated as such statistically.

In Experiment 3 the data from conditioning and testing were analyzed separately. The ANOVA for conditioning stage was 3 x 3 since there were three conditioning trials, and the ANOVA for extinction was 6 x 3 because it comprised six trials.

 3) Discussion: There are a number of issues concerning the interpretation of Exp 3 results. First, I’m not convinced that the results of Exp 3 say anything about anxiety. It is possible (probable) that VPA administered prior to fear conditioning would have a direct effect on the acquisition of fear, irrespective of any potential effect on anxiety. Second, it is necessary to rule out the (strong) possibility of state-dependent learning effects in Exp 3, especially since VPA has clearly demonstrated unconditioned motoric effects.

We agree with the reviewer that there are other possible interpretations for the reduced conditioning observe in the VAL/Before group in Experiment 3. In fact, we recognized such possibility in the first version of the ms when saying “we cannot completely rule out an account in terms of the possible effect of the drug on the perceived intensity of the shock, which could have weakened the strength of the association with the tone”. In addition, the hypothesis on the anxiolytic effect of Valproate is supported not only by the results of Experiment 3, but it has been repeatedly informed in previous research (e.g., Simiand et. al., 1984; De Angelis, 1992, 1995), that adds validity to our interpretation. 

Anyway, the paragraph on page 21 (lines 513-516) has been modified to reflect the reviewer’s suggestion, and now reads: “The reduction in fear conditioning for the group that received VPA before the tone-shock pairings in Experiment 3 was probably due to the unconditioned anxiolytic effect of VPA, a property that has been demonstrated in previous research using different anxiety behavioral task (e.g., De Angelis, 19-22).” 

Regarding a possible state dependent learning in Experiment 3, we think it is a really interesting topic that deserve additional research. However, in most experiments using drugs as the US and context as CS there is a minimum of 4 context – drug long pairings to support conditioning. Considering that in our Experiment 3 there were only one drug-context coincidence, it is unlikely that it supported a conditioned reduction of locomotor activity during the free-drug test. In the same vein, and taking into account that VPA produced a reduction in motor activity during the conditioning trial, we would have to interpret the increase in locomotor activity as an opponent CR that is not common when analyzing locomotor conditioning with other drugs that induce hypolocomotion.

  4) Generally, I believe this set of experiments, while conducted well, are inconclusive with regards to the overall interpretation of VPA effects on anxiety. Assessment of VPA effects on anxiety using a classic anxiety behavioral task (or multiple tasks) would significantly strengthen the interpretation the authors favor.

We agree with the reviewer that adding additional experimental evidence would strengthen our hypothesis about the anxiolytic properties of the VPA, but the existence of previous research showing such anxiolytic properties (that was already mentioned in the original ms, and that is now more deeply discussed in the revised version – see lines 598-614) can also be considered as strong support for our interpretation.   Minor concerns: 1) Lines 148-149: List groups in same order (in text) as listed in Table 1.

Done. In addition, we have also changed the term ‘Val’ for ‘VPA’ in the Figures and in the text to homogenize the groups’ names across all experiments. 

2) Clarify whether Figure 1B represents total consumption (minutes 1-60) or consumption for minutes 30-60.

Figure 1B represents total consumption (minutes 1-60). It has been clarified in the first paragraph of the results section for Experiment 1 (p. 9, lines 204, 210 and 214), and in the Figure 1 caption.

 3) Line 411: change “free-drug” to “drug-free”

Done

 4) It would be useful to include a direct assessment of footshock reactivity during acquisition in Exp 3 (e.g., activity burst; Fanselow, 1982). Fanselow, Michael S. “The Postshock Activity Burst.” Animal Learning & Behavior, vol. 10, no. 4, Nov. 1982, pp. 448–54.

We agree with the reviewer that including a direct assessment of footshock reactivity would be very useful for a better understanding of fear conditioning but, unfortunately, we only registered percent of locomotor activity for the complete ITI and CS duration and, consequently, we cannot identify the specific duration of activity bursts or any other indications of footshock reactivity. We will introduce such index in our future research.

---

## [Decision Letter · Decision Letter 1]

19 Jun 2023

Unconditioned and Conditioned Anxiolytic Effects of Sodium Valproate on Flavor Neophobia and Fear Conditioning

PONE-D-22-32967R1

Dear Dr. De la Casa,

We’re pleased to inform you that your manuscript has been judged scientifically suitable for publication and will be formally accepted for publication once it meets all outstanding technical requirements.

Kind regards,

Prof. Dr. Dragan HRNCIC, MD, MSc, MBE, PhD

Academic Editor

PLOS ONE

Additional Editor Comments (optional):

Reviewers' comments:

Reviewer's Responses to Questions

**Comments to the Author**

1. If the authors have adequately addressed your comments raised in a previous round of review and you feel that this manuscript is now acceptable for publication, you may indicate that here to bypass the “Comments to the Author” section, enter your conflict of interest statement in the “Confidential to Editor” section, and submit your "Accept" recommendation.

Reviewer #1: All comments have been addressed

2. Is the manuscript technically sound, and do the data support the conclusions?

Reviewer #1: Yes

3. Has the statistical analysis been performed appropriately and rigorously? 

Reviewer #1: Yes

4. Have the authors made all data underlying the findings in their manuscript fully available?

Reviewer #1: Yes

5. Is the manuscript presented in an intelligible fashion and written in standard English?

Reviewer #1: Yes

6. Review Comments to the Author

Reviewer #1: (No Response)

7. PLOS authors have the option to publish the peer review history of their article (what does this mean?). If published, this will include your full peer review and any attached files.

Reviewer #1: No

---

## [Editor Report · Acceptance letter]

28 Jun 2023

PONE-D-22-32967R1 

Unconditioned and Conditioned Anxiolytic Effects of Sodium Valproate on Flavor Neophobia and Fear Conditioning 

Dear Dr. De la Casa:

I'm pleased to inform you that your manuscript has been deemed suitable for publication in PLOS ONE. Congratulations! Your manuscript is now with our production department. 

Kind regards, 

on behalf of

Professor Dragan Hrncic 

Academic Editor

PLOS ONE